# SlimGPT: Layer-wise Structured Pruning for Large Language Models

**Gui Ling, Ziyang Wang, Yuliang Yan**[*]**, Qingwen Liu**
Alibaba Group
{linggui.lg, shanyi.wzy, yuliang.yyl, xiangsheng.lqw}@alibaba-inc.com

## Abstract

Large language models (LLMs) have garnered significant attention for their remarkable capabilities across various domains, whose vast parameter scales present challenges for practical deployment. Structured pruning is an effective method to balance model performance with efficiency, but performance restoration under computational resource constraints is a principal challenge in pruning LLMs. Therefore, we present a low-cost and fast structured pruning method for LLMs named *SlimGPT* based on the Optimal Brain Surgeon framework. We propose Batched Greedy Pruning for rapid and near-optimal pruning, which enhances the accuracy of head-wise pruning error estimation through grouped Cholesky decomposition and improves the pruning efficiency of FFN via Dynamic Group Size, thereby achieving approximate local optimal pruning results within one hour. Besides, we explore the limitations of layer-wise pruning from the perspective of error accumulation and propose Incremental Pruning Ratio, a non-uniform pruning strategy to reduce performance degradation. Experimental results on the LLaMA benchmark show that SlimGPT outperforms other methods and achieves state-of-the-art results.

## 1   Introduction

Large Language Models (LLMs) [1, 2, 3] have made significant strides in various natural language processing tasks, leading to the emergence of novel applications such as AI agents [4]. One of the factors contributing to the exceptional capabilities of LLMs is their massive parameter scales. However, these extensive parameters also introduce increased inference costs and deployment challenges, hindering the widespread application and adoption of LLMs. Accelerating inference for LLMs has become a focal point of current research. Model compression [5], as one of the strategies for inference acceleration, including techniques like pruning and quantization [6, 7], has been extensively researched. Nevertheless, earlier model compression techniques, particularly model pruning, typically rely on heavy post-training to recover the model's capabilities, which typically involves retraining with the entire training dataset. Given the constraints of current computational resources, the above approaches are not feasible for LLMs.

In the domain of LLM pruning, recent studies have largely focused on unstructured (or semi-structured) pruning [8], a method that shrinks models by selectively zeroing out weights considered non-critical. Despite its advancements, unstructured pruning falls short in substantially reducing parameter count, which is crucial for accelerating LLM inference as it is often bottlenecked on memory bandwidth and communication [9]. To accelerate inference speed, unstructured pruning models are often paired with specialized frameworks or hardware solutions. Conversely, *structured pruning* [10, 11] effectively decreases the model's parameter count by systematically eliminating columns or rows from weight matrices, enabling significant improvements in inference speed, and

---

[*]Corresponding author

reduce deployment cost on conventional hardware. Yet, structured pruning often entails more pronounced compromises in model performance, which poses a greater challenge.

Recently, researchers have applied the classic Optimal Brain Surgeon (OBS) framework to the compression of LLMs. This approach includes parameter compensation which can mitigate the loss incurred during compression and reduce the dependence on post-training. The OBS framework is currently applied in the areas of unstructured pruning [12] and quantization [13] for LLMs. However, there exist some challenges in its application to structured pruning:

- The OBS is a fine-grained compression framework that compresses one parameter at each iteration, whereas structured pruning has a minimum granularity of either a column or head. Directly applying the OBS framework will result in high numerical errors, impairing model performance.
- The OBS is essentially a layer-wise compression method. It focuses on each individual layer, thus failing to allocate pruning ratios for each layer rationally using global information (such as global gradients). This is crucial for LLM structured pruning, which relies on a non-uniform strategy to reduce the impact on performance.

To address these issues, we propose a new structured pruning method for LLMs. We introduce Batched Greedy Pruning to achieve low-cost and rapid pruning for LLMs. Specifically, for attention heads, we propose grouped Cholesky decomposition to select nearly optimal heads for pruning in each iteration, thereby maintaining an approximately locally optimal pruning result. For Feed-Forward Networks (FFNs), we achieve near-optimal and efficient pruning results through Dynamic Group Size. Furthermore, since the OBS is essentially a layer-wise compression framework, we investigate the error accumulation phenomenon in layer-wise pruning and propose pruning by Incremental Pruning Ratio, a straightforward non-uniform strategy to control the pruning rate of each layer, further mitigating performance loss under a given overall pruning ratio.

**Contribution.** In this paper, we propose SlimGPT, a layer-wise pruning approach that extends the classical OBS framework to structured pruning for LLMs. The characteristics of SlimGPT can be summarized as follows: **(i)** Task-agnostic pruning scheme. Only a random sample of data from generic pre-training corpora is needed as a calibration set, and we can obtain a compressed model with most performance preserved; **(ii)** Low-cost, low-resource, and time-efficient compression scheme. The model can be compressed using just a single GPU, a few hundred of calibration data, and about one hour; **(iii)** A universal pruning method for Transformer-based models. It has good transferability and, theoretically, is applicable to all large models based on the conventional Transformer architecture. We employ LLaMA models for pruning and conduct evaluations on wikitext2 and Commonsense Reasoning tasks. The results indicate that SlimGPT substantially retains the performance of the pruned models, surpassing state-of-the-art methods.

## 2    Related Work

**Compression methods with regularization**. Before the era of LLMs, using the scaling factors from Batch Normalization layers as indicators of channel importance made pruning based on regularization a very popular method [14, 15]. Notably, Louizos et al. [16] implemented the non-differentiable L0 penalty in a differentiable form, a technique frequently used for pruning in large models. Compresso [17] combines L0 regularization with LoRA training [18], effectively preserving model performance at a low cost. In a similar vein, Sheared LLaMA [19] employs augmented L0 regularization on inserted masks for structured pruning, using extensive data to restore performance and deliver compact yet powerful pruned models.

**Global gradient-based compression methods**. NVIDIA's works [20, 21] involve a Taylor expansion of the global loss. By eliminating higher-order terms, it is revealed that the impact of a weight on the loss can be assessed using the magnitude of the weight combined with gradient information. Based on this, LLM-Pruner [11] employs a first-order importance estimation to gauge the importance of weights. LORAPrune [22] measures the importance of weights based on the gradients of the LORA parameters rather than the model's parameters, achieving commendable results.

**Outliers-dependent compression methods**. Dettmers et al. [23] identifies an attribute unique to LLMs, where a small subset of activation values in the data features have magnitudes significantly larger than the others. And removing corresponding weights impacts model performance substantially. Building upon this, Wanda [24] proposes a simple yet effective unstructured pruning method, using

the product of a weight's L1 norm and the L2 norm of eigenvalues to gauge its importance, achieving impressive pruning results. OWL [25] determines layer-wise sparsity ratios based on Layerwise Outlier Distribution (LOD), obtaining substantial performance gains at high sparsity levels.

**Layer-wise compression methods**. The early works [26, 27] provide a layer-wise compression framework with a locally optimal solution named Optimal Brain Surgeon (OBS). And then OBC [28] reduces the computational burden by converting layer-wise pruning into row-wise pruning and updating the inverse Hessian using a proposed formula. Furthermore, GPTQ [13] accelerates the process with Lazy Batch-Updates and Cholesky Reformulation, enabling the application of this method to the quantization of LLMs. SparseGPT [12] also adapts this approach for unstructured pruning of LLMs. However, there appears to be no existing research that has implemented OBS in structured pruning for LLMs.

**Structured Pruning vs. Other Techniques**. Given that OBS has previously been used in both quantization and unstructured pruning, and is now being applied to structured pruning, there is an inherent consistency across these three compression schemes. These methods actually compress the model at varying levels of granularity. Quantization, which "trims" floating-point precision, represents the finest granularity and delivers excellent compression outcomes. Structured pruning, on the other hand, involves trimming weight vectors and represents the coarsest granularity, naturally resulting in higher performance losses compared to other methods, which poses significant challenges. For small models, it is possible to recover most of the performance with post-training, but this is challenging to achieve in LLMs due to resource constraints. Nonetheless, structured pruning effectively reduces the number of parameters without needing special inference framework support and is compatible with the other two methods, thus still holding considerable potential for application.

## 3 Preliminary

**Layer-Wise Pruning.** Consider the scenario of pruning on a well-optimized model, known as post-training pruning, a prevalent approach involves decomposing the global model pruning challenge into layer-wise subproblems (*i.e.,* Layer-wise pruning), which are typically modeled as issues of minimizing L2 error. Specifically, let $\mathbf{W}_l$ represent the weights of the $l$-th layer of a pretrained model and $X_l$ be the input features for layer $l$. The goal is to determine pruned weights $\hat{\mathbf{W}}_l$ that achieve a predefined pruning ratio while minimizing the squared error:

$$\text{argmin}_{\hat{\mathbf{W}}_1} \|\mathbf{W}_l X_l - \hat{\mathbf{W}}_l X_l\|_2^2. \tag{1}$$

**Optimal Brain Surgeon (OBS) Framework.** As Equation 1 can be rewritten as the sum of square error of each row of the weights to be pruned, the layer-wise pruning can be further split into row-wise pruning [28]. Consider the removal of a single weight from a row in $W_l$, Equation 1 has a closed-form solution [27]. Let $w$ denote a specific weight in a row of $W_l$, and let $p$ be its corresponding index. Given that our optimization objective is to minimize row-wise squared error, the Hessian of this objective with respect to the weight row of layer $l$ is given by $H_l = 2X_l X_l^T$. The weight to be pruned, $w_p$, as well as the necessary update $\delta_p$ applied to the remaining weights of the same row to counterbalance the removal, can be determined through the following calculation:

$$w_p = \text{argmin}_{w_p} \frac{w_p^2}{H_{p,p}^{-1}}, \quad \delta_p = -\frac{w_p}{H_{p,p}^{-1}} \cdot H_{:,p}^{-1}, \tag{2}$$

where $H_{p,p}^{-1}$ denotes the $p$ th diagonal entry of the inverse Hessian, and $H_{:,p}^{-1}$ is its $p$ th column. By iteratively using Equation 2 to remove one weight and update the remaining weights in the same row, one can obtain a locally optimal compressed model. After each iteration, $H$ will be updated by removing the $p$ row and column, which is represented by $H_{[-p]}$, here we use $[-p]$ to indicate the removal of $p$ row and column of the matrix. As $H^{-1}$ cannot be updated by simple removal as $(H_{[-p]})^{-1} \neq (H^{-1})_{[-p]}$, to avoid the expensive full recomputations of $H^{-1}$, the following formula is proposed to quickly update $H^{-1}$ [28]:

$$(H_{[-p]})^{-1} = (H^{-1} - \frac{1}{H_{p,p}^{-1}} H_{:,p}^{-1} H_{p,:}^{-1})_{[-p]}. \tag{3}$$

This framework can be practically applied to medium-sized models. However, for models with billions of weights, the iterative pruning becomes exceedingly time-consuming.

# 4 Methodology

In this section, by extending the OBS framework to structured pruning, we introduce SlimGPT from two aspects: (1) By employing **Batched Greedy Pruning** to reduce error computation, we minimize the performance degradation caused by pruning while also accelerating the pruning speed; (2) By analyzing the limitation of layer-wise pruning from the perspective of error accumulation, we introduce **Incremental Pruning Ratio**, a non-uniform pruning strategy.

## 4.1 Structured Pruning with OBS Framework

As mentioned above, the pruning between different rows is independent, making it possible to prune all rows simultaneously [29]. We extend the OBS framework to structured column pruning, *i.e.,* pruning one column at a time and compensating the rest columns using the following formula:

$$W_{:,p} = \text{argmin}_{W_{:,p}} \frac{\sum W_{:,p}^2}{H_{p,p}^{-1}}, \quad \Delta = -\frac{W_{:,p}}{H_{p,p}^{-1}} \cdot H_{p,:}^{-1}, \tag{4}$$

where $H_{p,:}^{-1}$ denotes the $p$-th row of $H^{-1}$, and the obtained $\Delta$ is a compensation matrix of the same size as $W$. We following previous works employ attention blocks and FFNs as the smallest units for pruning. By pruning the columns of the output matrix in attention blocks and the dimensionality reduction matrix in FFN blocks, we reduce the number of attention heads and FFN channels, thereby decreasing the model's parameter count.

However, the above formula cannot be applied directly, as iteratively finding and pruning the column with the minimum error is time-consuming. More critically, the structural dependency in attention blocks imposes additional constraints on column pruning, making it impossible to evaluate the importance of a head based solely on information from a single column.

## 4.2 Batched Greedy Pruning

Given that the calculation of the pruning error requires only the diagonal elements of $H^{-1}$ (see Equation 4), which are updated after each iteration, computing these elements in advance allows for calculating the head-wise error. With the observation that the sequential row removal via Equation 3 for the symmetric $H^{-1}$ essentially corresponds to taking a Cholesky decomposition [13], we can obtain the elements in advance with Cholesky decomposition.

Hoewever, the matrix obtained by Cholesky decomposition is triangular, and the elements of the current row (column) are calculated based on the elements of all the previous rows (columns), which means the Cholesky decomposition breaks the comparability between rows (columns). So it is hard to obtain all the required information in advance through the Cholesky decomposition like [12, 13], whose error comparison is usually within the same column but structured pruning requires the comparison of different columns.

Since structured pruning only requires traversing the columns that need to be removed, by rearranging the rows and columns corresponding to a head that is to be pruned in $H$ to the front, and then invert the matrix followed by Cholesky decomposition, we can calculate the

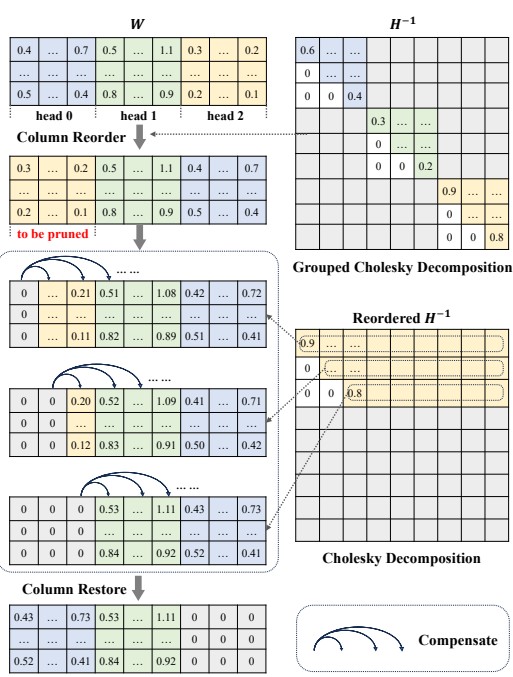

Figure 1: The figure illustrates Batched Greedy Pruning on attention blocks, where $W$ is a output matrix and $H$ is the corresponding Hessian. Different colors represent distinct attention heads and gray indicates the pruned weights.

**Algorithm 1** *Batched Greedy Pruning* for Attention Heads Given Weight matrix $W$, inverse Hessian $H^{-1}$, head size $d$ and head count $n$.

---

*// Step 1: calculate head-wise error*
$\hat{\mathbf{H}}^{-1} \leftarrow \text{GroupedCholesky}(\mathbf{H}^{-1})$
$\mathbf{E} \leftarrow \mathbf{W}^2/\text{Diag}(\hat{\mathbf{H}}^{-1})^2$    *// error matrix*
$\mathbf{E} \leftarrow [\sum \mathbf{E}_{:,0:d}, \sum \mathbf{E}_{:,d:2d}, ..., \sum \mathbf{E}_{:,(n-1)d:nd}]$    *// head error*
$\mathbf{A} \leftarrow \text{Head2ColumnIdx}(\text{Argsort}(\mathbf{E}))$    *// rerodered column index*
$\mathbf{W} \leftarrow \mathbf{W}[:, \mathbf{A}], \mathbf{H}^{-1} \leftarrow \mathbf{H}^{-1}[\mathbf{A}, :][:, \mathbf{A}]$    *// reorder*

*// Step 2: prune a head column-wise*
$\hat{\mathbf{H}}^{-1} \leftarrow \text{Cholesky}(\mathbf{H}^{-1})^T[:d, :]$
$\mathbf{E} \leftarrow \mathbf{0}_{d_{row} \times d}$
**for** $i$ in $0, 1, 2..d$ **do**
     $\mathbf{E}_{:,i:i+1} \leftarrow \mathbf{W}_{:,i:i+1}/\hat{\mathbf{H}}^{-1}_{i,i}$    *// pruning error*
     $\mathbf{W}_{:,i:d} \leftarrow \mathbf{W}_{:,i:d} - \mathbf{E}_{:,i:i+1} \times \hat{\mathbf{H}}^{-1}_{i,i:d}$    *// local update, column $i$ is zeroed*
**end for**
$\mathbf{W}_{:,d:} \leftarrow \mathbf{W}_{:,d:} - \mathbf{E} \times \hat{\mathbf{H}}^{-1}_{:,d:}$    *// global update*
$\mathbf{W} \leftarrow \mathbf{W}[:, \text{Argsort}(\mathbf{A})]$    *// restore*

---

head error column-wise. However, repeated rearrangement followed by matrix inversion and Cholesky decomposition is highly time-consuming, and this is just to find one head to be pruned.

We accelerate the above process through two common lemmas (proofs are provided in the Appendix): **(i)** For symmetric $H$, the inverse matrix after permutation can be obtained by the same permutation of $H^{-1}$; **(ii)** The principal submatrix of symmetric $H^{-1}$ after Cholesky decomposition is equivalent to the Cholesky decomposition of its principal submatrix. Thus we can calculate the pruning error of all the heads at once through *grouped Cholesky decomposition*. Specifically, we inverse $H$ once and split it into $n_{head}$ matrices along the main diagonal, with each remains definite and symmetric, and decompose them in parallel:

$$\hat{\mathbf{H}}^{-1} = \text{Cholesky}(\text{Stack}([H^{-1}_{0:d,0:d}, H^{-1}_{d:2d,d:2d}, ..., H^{-1}_{(n-1)d:nd,(n-1)d:nd}])) \quad (5)$$

where decomposed $\hat{\mathbf{H}}^{-1}$ is a matrix of size $n_{head} \times d_{head} \times d_{head}$, $n_{head}$ and $d_{head}$ represent the head number and head dimension, respectively. Utilizing GPU acceleration, we can quickly calculate the value of the diagonal element in advance and calculate the head-wise error. Note that during error computation, we only update the diagonal elements of $H^{-1}$ and skip the update of $W$, which is small and does not dominate the ordering of errors.

After determining the head to be pruned, we rearrange the corresponding columns of $W$ and the corresponding rows and columns of $H^{-1}$ to the front, and again use the global Cholesky decomposition on reordered $H^{-1}$ to prune the head column by column until the first head is pruned. In this way, we can avoid traversing columns that do not need pruning and only traverse necessary columns to improve pruning efficiency further. Figure 1 shows the process of Batched Greedy Pruning applied to attention blocks, and Algorithm 1 is a pseudocode illustrating how to prune a head with two steps: calculating head-wise error and pruning a head column-wise.

For FFNs, since there is no block constraint similar to attention heads, we can achieve local numerical optimality by pruning columns individually using Equation 4. However, the column-wise pruning is time-consuming because of the substantial intermediate dimensions of FFN. We thus prune a group of columns at a time and select the top-k columns with the most minor errors for pruning at each iteration. Considering that the compensation at each iteration may lead to a local reshuffling of column errors, we adopt a dynamic grouping strategy for pruning FFN blocks. We start with larger group size such as 1024 for pruning and gradually decrease the group size to a small number like 8, which allows us to enhance pruning efficiency while approaching an approximate optimal solution.

## 4.3 Incremental Pruning Ratio

Through Batched Greedy Pruning, we can obtain near-optimal structured pruning results for each layer. However, finding a suitable pruning ratio for each layer is difficult, as considering global information is quite challenging for layer-wise pruning, which only provides optimal pruning results

for the current layer. Maintaining a uniform pruning ratio across all layers is unreasonable and will impact model performance, especially when the pruning ratio is high. Existing works have different approaches to the problem. For example, LLM-Pruner [11] avoids pruning in the initial and final layers while maintaining a consistent ratio in the intermediate layers to manually implement non-uniform pruning. OWL [25] adjusts sparse ratios dynamically for each layer based on the proportion of feature outliers, which is applied to unstructured pruning.

We find that layer-wise pruning, particularly structured layer-wise pruning, suffers from error accumulation due to its locality. Errors introduced during pruning in one layer can be amplified in subsequent layers, resulting in significant discrepancies between the final model output and the original. Figure 2 presents the per-layer output error of FFN between the original model and three distinct pruned models. The pruned models each implement a first-layer pruning of 25%, 50%, and 75%, respectively. The error increases with model depth and accumulates at a rate exceeding linear progression as the initial layer's pruning ratio increases. Based on this observation, we propose a straightforward pruning strategy for layer-wise pruning, termed Incremental Pruning Ratio, which can effectively minimize pruning losses without any additional operation.

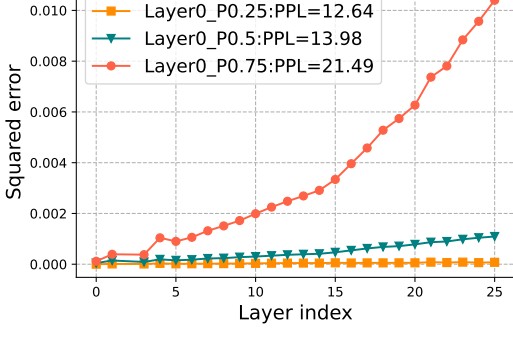

Figure 2: Per-layer FFN output error between the original LLaMA-7B and three distinct pruned models. The pruned models each implement a first-layer reduction of 25%, 50%, and 75%, respectively. The PPL of original model is 12.63. For ease of visualization, the layer index has been truncated to 25.

In Incremental Pruning Ratio, without loss of generality, we employ a logarithmically increasing strategy to control the layer-wise pruning ratio. Specifically, for an $n$-layer model with the first and last layer pruning ratios denoted as $r_0$ and $r_{n-1}$ respectively, the pruning ratio for the $i$-th layer is defined as follows:

$$r_i = r_0 + (r_{n-1} - r_0)\frac{\log(i+1)}{\log(n)}, \ (0 \leq i < n) \tag{6}$$

where $r_i$ represents the pruning ratio for the $i$-th layer. This formula ensures that the pruning ratio from the first layer to the last layer transitions smoothly as a logarithmic curve. The strategy mitigates the pruning error accumulation in shallow layers while avoiding the issue of excessive pruning in the deeper layers, allowing for further reduction in performance loss.

## 5 Experiment

### 5.1 Experimental Settings

**Implementation details.** We use C4 dataset [30] as the calibration set. From the first shard of C4, we randomly select 256 2048-token sequences for pruning. To restore performance, we following LLM-Pruner [11] finetune the pruned model with LORA [18]. We tune with Alpaca datsets [31] for one epoch and utilize the AdamW optimizer with an initial learning rate set to 1e-4, coupled with a cosine annealing schedule for the learning rate. The global batch size is set to 64 and the sequence length is truncated to 256. All pruning experiments are conducted on a single A100, while finetuning is performed using two A100s.

**Models and Metrics.** To assess the effectiveness and generality of SlimGPT, We carry out a series of experiments on the LLaMA families [2]. And to measure the effectiveness of our pruned models in the task-agnostic setting, we follow previous pruning works to evaluate language modeling performance and commonsense reasoning capabilities. The language modeling performance is evaluated on the WikiText2 [32] validation set with sequence length truncated to 128, and the commonsense reasoning capabilities is carried out under a zero-shot setting on the Commonsense Reasoning datasets, which encompass seven diverse subtasks: BoolQ [33], PIQA [34], HellaSwag [35], WinoGrande [36], ARC-easy [37], ARC-challenge [37], and OpenbookQA [38]. We utilize the lm-eval-harness framework [39] to conduct these evaluations.

Table 1: PPL & Commonsense Reasoning zero-shot performance of the pruned LLaMA-7B. The average score is computed across seven datasets. The **bolded** results represent the optimal results, while the underlined ones is the sub-optimal results. The asterisk-marked (*) results are those replicated within a consistent experimental framework, which slightly differ from the original source.

| Prune% | Method | #Params | PPL↓ | BoolQ | PIQA | HellaS | WinoG | ARC-e | ARC-c | OBQA | Avg. |
|---|---|---|---|---|---|---|---|---|---|---|---|
| - | -* | 6.7B | 12.63 | 75.08 | 79.16 | 76.20 | 70.00 | 72.89 | 44.88 | 44.40 | 66.09 |
| 20% | LLM-Pruner* | | 18.01 | 66.76 | 78.45 | 71.44 | 63.77 | 66.41 | 39.85 | 43.80 | 61.50 |
| | Compresso | | - | **79.08** | 75.46 | 53.44 | 67.80 | 68.64 | 37.97 | 34.20 | 59.51 |
| | LoraPrune | 5.4B | 16.80 | 65.62 | 79.31 | 70.00 | 62.76 | 65.87 | 37.69 | 39.14 | 60.06 |
| | SlimGPT w/o tune | | 16.99 | 75.93 | 77.58 | 73.07 | 67.96 | 68.60 | 41.72 | 41.80 | 63.81 |
| | SlimGPT | | **16.68** | 74.59 | 78.94 | **74.40** | **68.43** | 70.50 | 43.26 | 45.40 | 65.07 |
| 25% | LLM-Pruner* | | 20.57 | 62.81 | 76.93 | 69.21 | 60.46 | 63.34 | 38.14 | 39.80 | 58.67 |
| | Compresso | 5.0B | - | 73.55 | 73.07 | 49.16 | 64.80 | 66.20 | 37.20 | 29.80 | 56.25 |
| | SlimGPT w/o tune | | 19.11 | **75.11** | 76.77 | 70.60 | **67.25** | 66.75 | 40.40 | 40.40 | **62.47** |
| | SlimGPT | | **18.45** | 73.46 | **77.42** | 72.07 | 65.51 | **67.17** | 41.13 | 40.40 | 62.45 |
| 33% | LLM-Pruner* | | 24.50 | 62.02 | 74.92 | 64.41 | 61.80 | 53.79 | 32.00 | 38.80 | 55.39 |
| | Compresso | 4.5B | - | 68.69 | 72.85 | 47.18 | 63.38 | **65.99** | 35.07 | 29.00 | 54.59 |
| | SlimGPT w/o tune | | 24.55 | **72.72** | 75.68 | 68.10 | **66.54** | 62.29 | 37.03 | 40.20 | 60.37 |
| | SlimGPT | | **22.43** | 71.53 | **76.66** | **70.55** | 66.06 | 64.35 | **39.33** | **41.40** | **61.41** |
| 50% | LLM-Pruner* | | 40.64 | 60.21 | 68.88 | 47.86 | 54.62 | 43.94 | 27.73 | 35.20 | 48.35 |
| | LoraPrune | 3.4B | **30.12** | 61.88 | 71.53 | 47.86 | 55.01 | 45.13 | 31.62 | 34.98 | 49.72 |
| | SlimGPT w/o tune | | 38.83 | **65.87** | 70.35 | 54.62 | **59.59** | 49.71 | 31.06 | 34.40 | 52.23 |
| | SlimGPT | | 31.07 | 65.11 | **71.60** | **59.94** | 59.27 | **53.37** | **31.83** | 35.20 | **53.76** |

To validate the universality of SlimGPT, we conduct experiments on additional models and supplementary evaluation datasets. The results of these experiments can be found in the Appendix. We conduct further pruning experiments on Vicuna [40], LLaMA2 [41], and Baichuan [42], which yield results consistent with those observed using the LLaMA model. In addition, we engage in preliminary evaluations on more complex tasks, specifically MMLU [43] and LongBench [44]. Although SlimGPT exhibits slightly larger performance losses on these datasets, it still retains a significant advantage over the baseline models.

**Baselines.** We compare SlimGPT with the following recent SOTA works on structured pruning, which we could find during our experiments:

- LLM-Pruner [11], a gradient-based pruning approach, serves as our benchmark. This method involves a two-step process: a one-shot pruning followed by performance restoration through LORA fine-tuning.
- Compresso [17] is a pruning method based on sparse training, applying L0 penalty to manually inserted masks during the LORA fine-tuning phase and employing a cubic sparsity schedule to iteratively prune the model until the desired pruning ratio is achieved.
- LoRAPrune [22] utilizes gradients from the LORA module's parameters to determine the importance of the original model's parameters, thus requiring only gradient information from the LORA module, which significantly reduces computational demands.

## 5.2 Main Result

### 5.2.1 Performance Evaluation

To facilitate a more effective comparison of the evaluated results with prior works, we prune the LLaMA-7B model using four distinct pruning ratios—20%, 25%, 33%, and 50%—resulting in four smaller models with parameter counts of 5.4B, 5B, 4.5B, and 3.4B, respectively. Table 1 shows the detailed perplexity and zero-shot performance of pruned LLaMA-7B with four different sizes. Compared to other approaches, SlimGPT demonstrates superior performance in language modeling and commonsense reasoning across most subtasks. Under a pruning condition of 20%, SlimGPT achieves a slightly better perplexity score than the best existing results (16.68 vs. 16.80) and shows

Table 2: PPL & Commonsense Reasoning zero-shot performance of the pruned LLaMA-13B/30B. The perplexity is evaluated on Wikitext2 and the zero-shot average is computed across seven Commonsense Reasoning datasets. The **bolded** results represent the optimal results. The asterisk-marked (*) results are those replicated within a consistent experimental framework, which slightly differ from the original source. Detailed results are available in the Appendix.

| Prune% | Method | #Params | PPL↓ | Zero-shot Avg.↑ | #Params | PPL↓ | Zero-shot Avg.↑ |
|---|---|---|---|---|---|---|---|
| - | -* | 13.0B | 11.58 | 68.16 | 32.5B | 9.78 | 71.92 |
| 20% | LLM-Pruner* | | 16.62 | 65.68 | | 12.06 | 69.99 |
| | SlimGPT w/o tune | 10.4B | 14.87 | 66.37 | 26.0B | **11.59** | 71.13 |
| | SlimGPT | | **14.73** | **68.06** | | 11.69 | **72.56** |
| 50% | LLM-Pruner* | | 74.62 | 53.22 | | 22.33 | 59.47 |
| | SlimGPT w/o tune | 6.5B | 31.05 | 57.82 | 16.3B | 18.61 | 65.50 |
| | SlimGPT | | **26.38** | **59.49** | | **17.17** | **66.79** |

Table 3: Pruning Runtime and Memory Usage

| #Params | Runtime (20%) | Runtime (50%) | Memory |
|---|---|---|---|
| 7b | 678s | 1074s | 7375M |
| 13b | 1417s | 2475s | 11601M |

Table 4: Inference Latency and Memory Usage

| Prune% | #Params | Latency | Memory |
|---|---|---|---|
| - | 6.7B | 13.51ms | 27737MB |
| 20% | 5.4B | 11.89ms | 22497MB |
| 50% | 3.4B | 9.21ms | 14297MB |

a 3.6-point improvement in zero-shot average (65.07 vs. 61.50). As the pruning ratio increases to 50%, the advantages of SlimGPT become even more pronounced. SlimGPT without post-training represents an approximately 8% improvement over the baseline LLM-Pruner in average performance (52.23 vs. 48.35), and with post-training, the average performance improvement reaches up to 11% (53.76 vs. 48.35). Specifically, on a dataset like Hellaswag, the improvement soars up to 25% (59.94 vs. 47.86).

Moreover, we observe that although SlimGPT affects different subtasks to varying degrees, its impact is relatively balanced across different tasks, eliminating the occurrence of disproportionately large losses in particular tasks. At lower pruning ratios, some tasks such as BoolQ can even outperform the original unpruned model. Additionally, the effects of fine-tuning also differ among tasks, significantly improving tasks like HellaSwag and ARC-easy, while potentially causing negative side effects for tasks such as BoolQ and WinoGrande. This phenomenon is likely closely associated with the datasets used for fine-tuning.

For larger-scale models such as LLaMA-13B and LLaMA-30B, previous works have not provided pruning results for these models. Therefore, we solely compare our results to the LLM-Pruner baseline, concentrating on two specific pruning settings: a lower pruning ratio (20%) and a higher pruning ratio (50%). The replication of LLM-Pruner is consistent with the method described in the paper, where the pruned models by LLM-Pruner are finetuned with LORA.

Table 2 presents the pruning results of LLaMA-13B and LLaMA-30B, and we can draw similar conclusions: SlimGPT outperforms LLM-Pruner in terms of both PPL and zero-shot average scores even without post-training. Note that as the scale of the model increases, the performance loss due to pruning becomes smaller, suggesting a higher degree of parameter redundancy in larger models. At a low pruning ratio of 20%, the LLaMA-13B model's average performance in commonsense reasoning is nearly on par with that of the original, unpruned model (68.06 vs. 68.16). Similarly, the pruned LLaMA-30B model slightly outperforms the unpruned version (72.56 vs. 71.92). For the perplexity task, even though SlimGPT exhibits gaps compared to the original model, it still performs better than baseline, even at low pruning ratios.

Besides, we can find that the performance of LLaMA-13B pruned by 50% falls short compared to LLaMA-7B pruned by 20%. This highlights the limitations of low-cost fine-tuning, where resource constraints and training with techniques like LoRA result in limited performance recovery for the model. Therefore, using lower pruning ratios to compress smaller LLMs yields better returns.

Table 5: Pruning results under different strategies of SlimGPT. '-DGS' means removing Dynamic Group Size for FFN while '-GCD' means removing grouped Cholesky decomposition for attention blocks.

|  | PPL↓ | Zero-shot Avg.↑ |
|---|---|---|
| SlimGPT | 38.83 | 52.23 |
| - DGS | 39.73 (+0.90) | 51.63 (-0.60) |
| - GCD | 54.94 (+16.11) | 51.59 (-0.64) |

Table 6: Pruning results with different pruning ratio strategies.

|  | Model Size | PPL↓ | Zero-shot Avg.↑ |
|---|---|---|---|
| log increase (SlimGPT) | 3.40b | **38.83** | 52.23 |
| linear increase | 3.34b | 46.57 (+7.74) | **53.45** (+1.22) |
| uniform | 3.50b | 123.05 (+84.22) | 44.34 (-7.89) |
| log decrease | 3.40b | 380.69 (+341.86) | 36.73 (-15.50) |
| linear decrease | 3.34b | 932.64 (+893.81) | 35.62 (-16.61) |

### 5.2.2 Efficiency Analysis

The pruning runtime and memory usage for LLaMA-7B and LLaMA-13B are detailed in Table 3. Memory usage fluctuates based on the model size and the calibration scale, while the pruning speed is additionally affected by the pruning ratio. We demonstrate the pruning efficiency results derived from our experimental setup. Utilizing SlimGPT, which operates on a layer-wise basis, there is no need to load the entire model at once. Instead, we only load the parameters of the current layer along with the corresponding input features, significantly reducing memory consumption. For instance, to prune the 7B model by 20%, approximately 7 GB of GPU memory and 18 minutes are required to complete the process. Similarly, pruning the 13B model by 50% necessitates around 12 GB of GPU memory and 41 minutes to finalize.

Table 4 illustrates the inference latency and memory usage of the pruned LLaMA-7b models. We prune LLaMA-7b by 20% and 50% respectively. The maximum output limit is set to 512 and the presented values are the average derived from 50 inference trials. When pruning 50% of the parameters, the memory usage of the model during inference decreases to approximately 51% (14297MB vs. 27737MB), and the inference latency is reduced to about 69% (9.21ms vs. 13.51ms).

### 5.3 Ablation Study

We systematically analyze the influence of several key parts of SlimGPT on the pruning effect, including the Batched Greedy Pruning and Incremental Pruning Ratio strategy. Within the calibration dataset, we conduct thorough experiments with sample sizes and sequence lengths. Unless specifically stated otherwise, all the following experiments are conducted under the condition of pruning 50% of LLaMA-7b **without** further post-training, to eliminate potential confounding effects. Supplementary ablation experiments can be found in the Appendix.

### 5.3.1 Impact of Batched Greedy Pruning Strategy

We leverage grouped Cholesky decomposition to enhance the accuracy of head-wise error computation in attention blocks. Similarly, for FFNs, our proposed Dynamic Group Size substantially increases pruning efficiency while preserving near-optimal pruning results. To validate the effectiveness of these two strategies, we start with the complete SlimGPT algorithm and first remove the Dynamic Group Size (denoted as '-DGS'), setting the group size for FFN pruning to a fixed value of 128. Then, we remove the grouped Cholesky decomposition (denoted as '-GCD') and use the initial $H^{-1}$ to calculate head-wise errors. The experimental results are shown in Table 5. For attention blocks, the grouped Cholesky decomposition strategy plays a key role in language modeling capabilities by improving the accuracy of error compensation. Replacing it with ordinary Cholesky decomposition results in a significant increase in PPL (38.83 vs 54.94). In comparison to the naive fixed group size scheme for FFNs, the Dynamic Group Size strategy proposed contributes to maintaining the model's commonsense reasoning performance (52.23 vs 51.63).

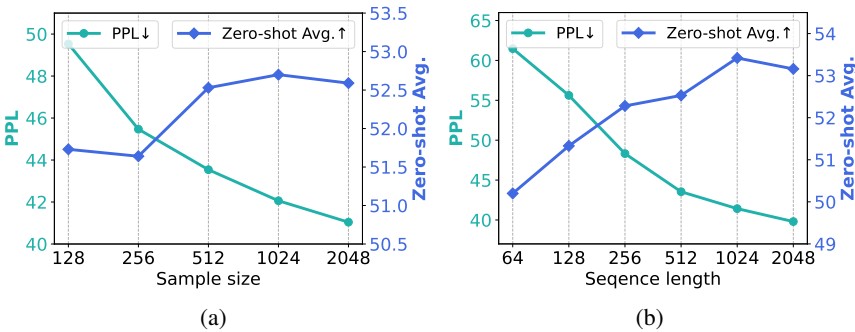

Figure 3: Effects of Calibration Sample Size & Sequence Length.

### 5.3.2 Impact of Incremental Pruning Ratio Strategy

The Incremental Pruning Ratio is a strategy specifically proposed for addressing the issue of layer-wise pruning. To maintain generality, we selected various common non-uniform strategies for comparative experiments, including logarithmic and linear increase strategies, as well as their corresponding decrease strategies. Among these, the logarithmic increase strategy is the default configuration for SlimGPT. Additionally, we conduct experiments under the setting of uniform pruning. Table 6 details the results under the different settings. From an overall perspective, the increase strategy for the pruning ratio has a clear advantage over uniform, and likewise, uniform shows a distinct advantage over decrease. Such results further verify the phenomenon of layer-wize error accumulation. As for the increase strategies of logarithmic and linear changes, due to disparities in model sizes, their results are not entirely comparable. The former performs best in language modeling (38.83), while the latter shows better performance in common sense reasoning tasks (53.45).

### 5.3.3 Effects of Calibration Samples & Sequence Length

We delve further into the impact of calibration samples and sequence length, and we choose C4 dataset for our experiments as it has a longer average sequence length. In exploring the effects of the sample scale, we fix the sequence length at 256 and test five scales ranging from 128 to 2048; similarly, when investigating the impact of sequence length, the sample scale is set to 256, with choices of sequence length varying from 64 to 2048. Figure 3 presents the perplexity result and zero-shot performance with different calibration samples and sequence lengths. As the number of samples increases, the PPL and zero-shot averages show a positive overall trend. Furthermore, after the sample count reaches 2048, the PPL does not bottom out, and there is room for further reduction. Similar phenomena can be observed in experiments on sequence length. With more sufficiently high-quality datasets with longer sequences, we believe SlimGPT can achieve better pruning effects.

## 6 Conclusion

In this work, we introduce a fast, structured pruning method for large-scale models within resource-constrained scenarios, based on the OBS framework, termed SlimGPT. Leveraging the novel Batched Greedy Pruning, we enhance the accuracy of pruning error estimation, thereby minimizing performance degradation from pruning. Moreover, we analyze the limitations of layer-wise pruning from the perspective of error accumulation and propose a non-uniform strategy named Incremental Pruning Ratio, which effectively improves the pruned model's performance. Evidence from open-source experiments affirms the efficacy of our approach.

**Limitations.** Even though SlimGPT achieves SOTA results in the structured pruning of LLMs, the model performance degradation at high pruning ratios (*e.g.,* 50%) or on more complex tasks (*e.g.,* LongBench) is still significant. How to enhance the model compression effectiveness under low-resource conditions remains a challenge. Moreover, we utilized a naive logarithmic change strategy in the Incremental Pruning Ratio, which, while ensuring generality, is not the optimal solution. The most suitable non-uniform approach requires further exploration. Lastly, similar to many large-scale open-source models available today, the model obtained through pruning by SlimGPT poses risks in terms of ethical safety and requires cautious handling.

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

# A Proof of Lemmas

## A.1 Proof of Lemma (i)

*Lemma.* For symmetric matrix $M$, the inverse matrix after permutation can be obtained by the same permutation of $M^{-1}$;

*Proof.* The lemma can be easily proven through elementary matrix transformations. Let $P$ be a permutation matrix. We have $P^T P = I$. And since $M$ is symmetric, $M = M^T$. We wish to prove that the inverse of the permuted matrix $M' = P^T M P$ is $(M')^{-1} = P^T M^{-1} P$. By the following transformations:

$$M'(P^T M^{-1} P) = (P^T M P)(P^T M^{-1} P) = P^T (M(PP^T))M^{-1}P = P^T MIH^{-1}P = I \quad (7)$$

we can demonstrates that $(M')^{-1} = P^T M^{-1} P$.

## A.2 Proof of Lemma (ii)

*Lemma.* The principal submatrix of symmetric $M$ after Cholesky decomposition is equivalent to the Cholesky decomposition of its principal submatrix.

*Proof.* Consider a symmetric matrix $M$. Without loss of generality, let's consider we are removing the last row and column. In block form:

$$M = \begin{bmatrix} A & B \\ B^T & C \end{bmatrix}, \quad (8)$$

its Cholesky decomposition can be expressed as:

$$M = LL^T = \begin{bmatrix} L_A & 0 \\ L_B & l \end{bmatrix} \begin{bmatrix} L_A^T & L_B^T \\ 0 & l \end{bmatrix}, \quad (9)$$

where $L_A$ is the Cholesky decomposition of $A$, and $l$ is a scalar value. Here, $A = L_A L_A^T$, and this matches the definition of the Cholesky decomposition for the principal submatrix $A$ of $M$. Thus the statement is demonstrated through the uniqueness of the Cholesky decomposition.

Table 7: PPL & Commonsense Reasoning zero-shot performance of the pruned LLaMA-13B/30B

| Prune% | Method | #Params | PPL↓ | BoolQ | PIQA | HellaS | WinoG | ARC-e | ARC-c | OBQA | Avg. |
|---|---|---|---|---|---|---|---|---|---|---|---|
| - | -* | 13.0B | 11.58 | 77.89 | 80.14 | 79.06 | 72.85 | 74.75 | 47.61 | 44.80 | 68.16 |
| 20% | LLM-Pruner* | | 16.62 | 79.38 | 77.36 | 71.47 | 70.32 | 70.54 | 44.88 | **45.80** | 65.68 |
| | SlimGPT w/o tune | 10.4B | 14.87 | 77.06 | 79.82 | 76.94 | 72.61 | 69.78 | 44.80 | 43.60 | 66.37 |
| | SlimGPT | | **14.73** | **80.00** | **80.47** | **78.44** | **72.69** | **71.59** | **47.61** | 45.60 | **68.06** |
| 50% | LLM-Pruner* | | 74.62 | 62.35 | 72.74 | 58.43 | 55.88 | 51.89 | 33.02 | **38.20** | 53.22 |
| | SlimGPT w/o tune | 6.5B | 31.05 | 69.14 | 74.32 | 64.57 | **65.82** | 57.74 | 35.15 | 38.00 | 57.82 |
| | SlimGPT | | **26.38** | **71.44** | **75.57** | **68.08** | 64.96 | **61.78** | **36.77** | 37.80 | **59.49** |
| - | -* | 32.5B | 9.78 | 82.69 | 82.26 | 82.60 | 75.85 | 78.91 | 52.90 | 48.20 | 71.92 |
| 20% | LLM-Pruner* | | 12.06 | 81.28 | 80.96 | 80.66 | 73.16 | 76.98 | 49.49 | 47.40 | 69.99 |
| | SlimGPT w/o tune | 26.0B | **11.59** | 82.87 | 81.28 | 81.01 | **76.09** | 76.98 | 51.28 | 48.40 | 71.13 |
| | SlimGPT | | 11.69 | **84.01** | **82.37** | **81.94** | 76.01 | **80.81** | **54.01** | **48.80** | **72.56** |
| 50% | LLM-Pruner* | | 22.33 | 66.21 | 76.44 | 69.46 | 64.56 | 60.98 | 37.63 | 41.00 | 59.47 |
| | SlimGPT w/o tune | 16.3B | 18.61 | 75.08 | 77.20 | 75.01 | **74.11** | 68.43 | 43.26 | 45.40 | 65.50 |
| | SlimGPT | | **17.17** | **75.93** | **77.91** | **77.43** | 73.80 | **70.62** | **44.45** | 47.40 | **66.79** |

# B More Detailed Evaluation Results

**Detailed evaluation results of pruned LLaMA-13B/30B.** Table 7 details the experimental results for LLaMA-13B/30B. The evaluation results in this table represent a detailed version of Table 2, listing scores for each specific commonsense task to provide a more detailed comparison.

Table 8: PPL & Commonsense Reasoning zero-shot performance of the pruned Vicuna-7B

| Prune% | Method | #Params | PPL↓ | BoolQ | PIQA | HellaS | WinoG | ARC-e | ARC-c | OBQA | Avg. |
|---|---|---|---|---|---|---|---|---|---|---|---|
| - | -* | 6.7B | 16.11 | 78.41 | 78.56 | 74.68 | 70.09 | 72.01 | 43.77 | 43.40 | 65.85 |
| 20% | LLM-Pruner* | 5.4B | 19.11 | 61.96 | 76.88 | 69.18 | 63.30 | 61.83 | 37.88 | 39.40 | 58.63 |
| | SlimGPT w/o tune | | 21.14 | **75.41** | 77.09 | **72.34** | **68.43** | **69.23** | 41.47 | **43.40** | 63.91 |
| | SlimGPT | | **17.73** | 74.98 | **77.42** | 72.19 | 67.88 | 68.31 | **41.47** | 42.60 | 63.55 |
| 50% | LLM-Pruner* | 3.4B | 43.96 | 40.76 | 67.08 | 46.64 | 53.28 | 43.98 | 27.56 | 34.00 | 44.76 |
| | SlimGPT w/o tune | | 42.90 | **65.84** | 71.22 | 54.08 | 56.83 | 54.71 | 31.40 | 35.60 | 52.81 |
| | SlimGPT | | **31.41** | 61.04 | **71.33** | **58.87** | **57.85** | **55.64** | **32.08** | **36.60** | **53.34** |

**PPL & Commonsense Reasoning evaluations of pruned Vicuna-7B.** Table 8 details the experimental results for Vicuna-7B. We observe that, on the Wikitext2 dataset, SlimGPT without finetuning exhibits comparable or higher PPL than LLM-Pruner, a result that diverges from findings in experiments with LLaMA models. The parameter compensation of SlimGPT makes it more dependent on the distribution of the calibration set compared to LLM-Pruner, while Vicuna is a model finetuned on general instructions, and at this point, pretrained data is not the most appropriate calibration set. Using an instruction dataset for pruning might yield better results, which remains to be verified. However, SlimGPT with finetuning still leads on most of the tasks.

Table 9: PPL & Commonsense Reasoning zero-shot performance of the pruned LLaMA2-7B

| Prune% | Method | #Params | PPL↓ | BoolQ | PIQA | HellaS | WinoG | ARC-e | ARC-c | OBQA | Avg. |
|---|---|---|---|---|---|---|---|---|---|---|---|
| - | -* | 6.7B | 12.19 | 77.71 | 79.05 | 76.00 | 68.98 | 74.58 | 46.33 | 44.20 | 66.69 |
| 20% | LLM-Pruner* | 5.4B | 17.00 | 67.95 | 77.58 | 71.43 | 64.01 | 63.51 | 38.05 | 39.80 | 60.33 |
| | SlimGPT w/o tune | | 16.49 | 73.43 | 77.58 | 72.62 | 68.82 | 69.99 | 42.32 | 42.00 | 63.82 |
| | SlimGPT | | **16.02** | **76.06** | **78.73** | **74.94** | **69.30** | **72.73** | **45.14** | **43.40** | **65.75** |

Table 10: MMLU 5-shot performance of the pruned LLaMA2-7b

| Prune% | Method | #Params | Humanities | Social Sciences | STEM | Other | Avg |
|---|---|---|---|---|---|---|---|
| - | -* | 6.7B | 43.3 | 51.6 | 36.3 | 52.1 | 45.6 |
| 20% | LLM-Pruner* | 5.4B | 25.7 | 23.6 | 24.2 | 26.8 | 25.2 |
| | SlimGPT w/o tune | | **36.0** | **45.2** | **33.5** | **44.1** | **39.4** |
| | SlimGPT | | 35.3 | 42.2 | 31.5 | 43.0 | 37.8 |

**PPL & Commonsense Reasoning & MMLU evaluations of pruned LLaMA2-7B.** LLaMA2-7B is a new generation model with completely different parameters, exhibiting better overall performance compared to the first generation LLaMA-7B. In addition to the Perplexity and Commonsense Reasoning assessments, we also supplement evaluation on the Massive Multitask Language Understanding (MMLU) task. MMLU is a quiz bank covering 57 subjects, presenting a greater challenge compared to the Commonsense Reasoning datasets. We evaluate using LLaMA2-7B with 20% of its parameters pruned, under 5-shot settings. The evaluation results for PPL and Commonsense Reasoning are shown in Table 9, while the results on the MMLU task are presented in Table 10. In the Commonsense Reasoning task, SlimGPT significantly outperforms the baseline and closely approaches the performance of the original unpruned model. In the MMLU task, although SlimGPT still substantially leads over the baseline, it exhibits a noticeable gap compared to the unpruned model and shows a slight decline after finetuning. For such challenging tasks, full post-training is required to restore performance, rather than relying solely on lightweight LoRA finetuning.

**PPL & Commonsense Reasoning & MMLU evaluations of pruned Baichuan-7B.** We conduct pruning experiments on the Baichuan-7b model and perform evaluations on the Wikitext2, Commonsense Reasoning datasets, and MMLU datasets. Tables 11 and Table 12 present the evaluation results for commonsense reasoning and MMLU, respectively. Similar to the findings with LLaMA2-7b, under the same LoRA finetuning settings, SlimGPT shows a clear improvement over the baseline.

Table 11: PPL & Commonsense Reasoning zero-shot performance of the pruned Baichuan-7B

| Prune% | Method | #Params | PPL↓ | BoolQ | PIQA | HellaS | WinoG | ARC-e | ARC-c | OBQA | Avg |
|---|---|---|---|---|---|---|---|---|---|---|---|
| - | -* | 7B | 13.25 | 70.09 | 76.93 | 70.06 | 64.09 | 67.05 | 40.53 | 38.20 | 60.99 |
| 20% | LLM-Pruner* | | 19.85 | 62.87 | 74.48 | 63.03 | 60.93 | 60.31 | 36.86 | 36.20 | 56.38 |
| | SlimGPT w/o tune | 5.7B | 20.01 | **69.17** | **75.03** | 65.25 | 61.25 | **64.94** | 35.15 | 36.60 | 58.20 |
| | SlimGPT | | **19.73** | 66.21 | **75.03** | **66.74** | **63.85** | 62.84 | **38.91** | **38.00** | **58.80** |

Table 12: MMLU 5-shot performance of the pruned Baichuan-7B

| Prune% | Method | #Params | Humanities | Social Sciences | STEM | Other | Avg |
|---|---|---|---|---|---|---|---|
| - | -* | 7B | 38.1 | 49.2 | 35.2 | 47.7 | 42.1 |
| 20% | LLM-Pruner* | | 24.8 | 23.4 | 21.9 | 26.8 | 24.3 |
| | SlimGPT w/o tune | 5.7B | 29.7 | 38.0 | **31.3** | 38.9 | 34.0 |
| | SlimGPT | | **32.2** | **39.2** | 31.1 | **40.2** | **35.4** |

Table 13: LongBench evaluation results of the pruned Mistral-7B-Instruct-V2.0

| Prune% | Method | #Params | Single-Doc QA | Multi-Doc QA | Summarization* | Few-shot |
|---|---|---|---|---|---|---|
| - | - | 6.7B | 35.8 | 29.4 | 24.2 | 65.8 |
| 20% | SlimGPT | 5.4B | 30.8 | 27.2 | 21.4 | 60.0 |

**Long Context Understanding evaluation results.** To further explore the impact of SlimGPT on the understanding of long-context texts, we select the Mistral-7B-Instruct-V2.0 model [45] for experiments, which supports up to 32k context windows. We prune it by 20% and conduct an evaluation on the LongBench task. Table 13 presents the evaluation results of the model before and after pruning. Note that we skip the evaluation of the GovReport datasets and thus the average score on Summarization tasks does not include that dataset. Under the LoRA finetuning settings, using SlimGPT with 20% of its parameters pruned can retain 90% of its long-text comprehension capabilities.

## C  Supplementary Ablation Experiments

### C.1  Influence of Calibration Data Category.

As SlimGPT updates the remaining parameters to mitigate the effects of pruning, which is dependent on the calibration data, it underscores the importance of investigating the impact of various calibration dataset categories. We conduct experiments on three general datasets:

- **C4 subset:** A commonly used pre-training corpus, which is the default calibration set for SlimGPT. We sample 512 sentences with 512 tokens from the first 20,000 corpus.

- **Alpaca dataset:** A high quality generic domain dataset used for supervised finetuning, generated by GPT3.5. We randomly sample 512 sentences with 512 tokens.

- **GPT4-Alpaca dataset:** A high quality dataset similar to Alpaca generated by GPT4. We randomly sample 512 sentences with 512 tokens.

We maintain consistency in pruning strategies across all models, differing only in the dataset used. Each model is pruned by 50% . We assess performance directly on these pruned models without any post-training. Table 14 presents the pruning results across various datasets. The three datasets can be categorized into pre-training datasets (C4) and instruction-following datasets (Alpaca, GPT4_Alpaca). Models pruned on C4 exhibit better PPL results on Wikitext2, whereas models pruned on Alpaca series perform better on the Commonsense Reasoning dataset. Different types of datasets have varying impacts on SlimGPT. Instruction-following datasets is more favorable for retaining the model's commonsense knowledge, whereas using pre-training datasets can achieve a balance between language modeling capabilities and commonsense abilities.

Table 14: Pruning results with various calibration datasets

| Dataset | PPL↓ | Zero-shot Avg.↑ |
|---|---|---|
| C4 (SlimGPT) | **42.06** | 52.70 |
| Alpaca | 48.26 | 54.44 |
| GPT4-Alpaca | 47.06 | **54.66** |

# D   More Analysis

## D.1   About Structural Dependency

The structural dependency problem in attention blocks happens when a column of weights in an attention head is removed, elements in other positions in the attention matrix are also affected because of the $softmax$ function. Directly summing the errors across all columns of a head may result in significant numerical inaccuracies, as Equation 4 applies only to single-column pruning instead of multiple columns. To achieve multi-column pruning, we need to iterate using Equation 4 and update $H^{-1}$ with Equation 3, which makes it difficult to assess the pruning error of a total attention head in advance.

## D.2   Layer-wise Pruning Ratio at Pruning Stage

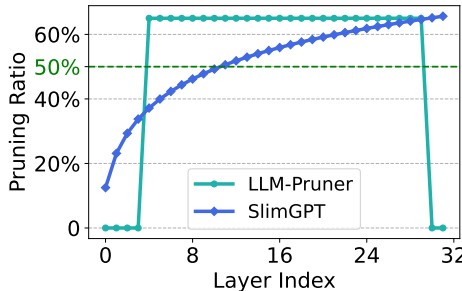

Figure 4: Layer-wise pruning ratio on LLaMA-7B with total pruning ratio 50%.

To more conveniently present the details of the logarithmic increase variation in Incremental Pruning Ratio, we illustrate the layer-wise pruning ratios for SlimGPT's logarithmic increase and LLM-Pruner's heuristic setting at a 50% pruning rate in Figure 4. SlimGPT starts with a lower initial pruning rate, with a rapid increase in the shallower layers followed by a slower change in the deeper layers, eventually approximating the fixed pruning ratio of LLM-Pruner. Their biggest difference lies in the handling of the last two layers. LLM-Pruner lacks parameter compensation, so the layers pruned at the output end have a larger impact on the final results, whereas SlimGPT reduces their impact on the model through parameter compensation.

## D.3   Training Loss at Recovery Stage

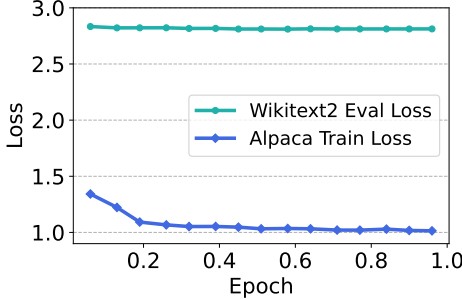

Figure 5: Alpaca train loss & Wikitext2 evaluation loss.

Table 15: Generated Examples from the LLaMA-7B and Pruned LLaMA-5.4B

| Model | Generated Sentences |
|---|---|
| LLaMA-7b | AI can create a logo in seconds, but it's not ready for the real world. By Dyllan Furness @dyllonline — Posted on July 12, 2017 - 9:35AM 07.12.17 - 9:35AM. A new artificial intelligence system has been developed that can generate logos from scratch with just one click of a mouse button. The technology is impressive and could have some practical applications down the line, but there are still many hurdles to overcome before we see this kind of software being used by designers or marketers. ... |
| | Neural Sequential Model, especially transformers, have been the most successful model in recent years. Their success is due to their ability to learn long-range dependencies and capture complex interactions between words. However, they are still far from perfect as they struggle with some of the challenges such as low performance on out-of-domain data or noisy labels. In this talk we will discuss how these models can be improved by using a combination of pretraining techniques ... |
| | What's great about the holiday season, is that it gives us a chance to reflect on what we are thankful for. I am so grateful and blessed with my family, friends, clients, colleagues, mentors, teachers, students... and all of you! I have been given an amazing opportunity in life to do something I love every day – help people find their voice through music. It has truly changed me as a person and helped me grow into who I want to be. Thank you for being part of this journey with me. |
| LLaMA-5.4b w/o tune | AI can create a logo in seconds, but it's not the same as having an expert design one for you. AI is great at creating logos that are visually appealing and look good on paper or screen. But if your company needs to have a unique brand identity, then there's no substitute for working with a professional designer who understands what makes a strong visual statement. |
| | Neural Sequential Model, especially transformers, have been widely used in many tasks. However, the performance of these models is still limited by their lack of generalization ability and high dependence on data. In this paper we propose a novel method to improve the generalizability of neural sequential model: using an auxiliary task as a pre-training step for the main task. We use a simple image classification problem with 10 classes (cars) as our auxiliary task. ... |
| | What's great about the holiday season, is that it gives us a chance to spend time with our family and friends. It also allows for some much needed downtime from work or school. But what happens when you have no one to celebrate with? Or maybe your loved ones are far away and can't make it home this year? Well, don't worry! There are plenty of ways to enjoy the holidays even if you aren't surrounded by people who love you as much as you do them. Here are 5 tips on how to survive being alone during Christmas: ... |
| LLaMA-5.4b w/ tune | AI can create a logo in seconds, but it's not the same as having an experienced designer. AI is great for creating logos and other visual elements that are simple or repetitive, like icons or illustrations. But when you need something more complex—like a custom design with unique typography, color schemes, and layout — human creativity comes into play. That said, there are some tools out there that make it easier to work with AI-generated designs: Adobe has released its own tool called Creative Intelligence (ACI), ... |
| | Neural Sequential Model, especially transformers, have been widely used in natural language processing (NLP) tasks. However, the performance of these models is still limited by their lack of robustness to noisy data and out-of-distribution samples. In this paper we propose a novel approach for improving the generalization ability of neural sequential model through incorporating adversarial training into the pretraining process. We show that our method can significantly improve the performance on various NLP tasks ... |
| | What's great about the holiday season, is that it gives us a chance to spend time with our loved ones. But what if you don't have any family or friends around? Don't worry! There are plenty of ways for you to enjoy your own little Christmas celebration and make this year even more memorable than ever before. Here are some ideas: 1) Have an intimate dinner party at home – Invite close friends over for a cozy evening in where everyone can share their favorite dishes and stories from the past year. ... 2) ... |

To figure out whether overfitting has occurred during the finetuning phase, potentially affecting the performance evaluation of the pruned models, we plot the loss curve of the model during the fine-tuning stage, as shown in Figure 5. We train for one epoch on the Alpaca dataset while using Wikitext2 as the evaluation set. The figure illustrates the train loss on Alpaca and the evaluation loss on Wikitext2. As is shown, the training loss is decreasing and converging normally, with no

significant fluctuations in the evaluation loss on Wikitext2, indicating that fine-tuning is conducted on general data without specific optimization for Wikitext2, and there is no occurrence of overfitting.

# E    Generation Cases from Pruned Model

Table 15 shows the generation cases of the original LLaMA-7B model, the pruned LLaMA-5.4B model, and the pruned and finetuned LLaMA-5.4B model. All inference parameters are kept consistent. To avoid data contamination from the fine-tuning process, we following LLM-Pruner select three input cases. From a qualitative analysis perspective, the model post-pruning by 20% shows little difference from the original LLaMA-7B. After fine-tuning, the model's output tends to offer suggestions more, likely due to the influence of the Alpaca dataset, but it still maintains a high standard in terms of generation quality.

