# OpenReview forum: "SlimGPT: Layer-wise Structured Pruning for Large Language Models"
_NeurIPS.cc/2024/Conference — NeurIPS 2024 poster_

### Official Review · Reviewer_8hG2 · 2024-07-09

**Soundness:** 4
**Presentation:** 4
**Contribution:** 4
**Rating:** 6
**Confidence:** 4

**Summary:**

This paper presents a novel SlimGPT framework to conduct structured pruning for LLMs in a fast and low-cost way. Specifically, SlimGPT modifies the Optimal Brain Surgeon (OBS) framework, and proposes a Batched Greedy Pruning to  enhance the performance of head-wise pruning through Cholesky decomposition. SlimGPT also improves the FFN pruning efficiency via Dynamic Group Size. Besides, SlimGPT employs an Incremental Pruning Ratio in order to mitigate the error accumulation problem in layer-wise pruning. Experiments on the LLaMA, LLaMA-2, and other popular LLMs demonstrate that SlimGPT achieves a new SOTA on LLM pruning.

**Strengths:**

1. This paper presents a proper way to extend the OBS framework to structured pruning with strong theoretical foundation. The technical details are thorough and convincing.
2. Extensive experiment results demonstrate that SlimGPT successfully achieves a new SOTA, surpassing all related works in this field.

**Weaknesses:**

1. SlimGPT employs an Incremental Pruning Ratio strategy. In L220, the article specifies that a logarithmic increasing strategy performs well and is employed in all experiments. Actually, this should be considered more carefully. Pruning can be viewed as a method to get rid of unnecessary information in the activations and only preserve necessary components for later layers. From this perspective, it resembles that of Token Merging [1,2]. It is already demonstrated in [1,2] that the performance loss caused by an aggressive pruning schedule in the first layers  can be mitigated by re-training. Therefore, I suggest the authors test more increasing strategies, instead of the logarithmic strategy, to fully utilize the power of SlimGPT.
2. Pruning at a $p\\%$ sparsity does not usually lead to an $\frac{1}{1-p\\%}$ throughput speedup [2]. To demonstrate that SlimGPT actually helps to deploy LLMs, the authors should carefully compare the **throughput** (eg., tokens per sec.) of pruned models generated by SlimGPT and the competing baselines.

[1] Token Merging: Your ViT But Faster (ICLR 2023)

[2] PYRA: Parallel Yielding Re-Activation for Training-Inference Efficient Task Adaptation (ECCV 2024)

**Questions:**

Under the same total sparsity value, does different increasing schedules affect the model inference speedup?

**Limitations:**

See Weaknesses and Questions. I do appreciate the theoretical contributions. So my rating may be adjusted after carefully checking the replies and other reviews.

---

> ### Author Rebuttal · Authors · 2024-08-06
>
> We truly appreciate the reviewer for the constructive comments.
>
> > W1: Further explanation about Incremental Pruning Ratio strategy.
>
> In Section 5.3.2, we have discussed the impact of different pruning ratio strategies on performance, with detailed results presented in Table 5 (for convenience, Table 5 is reproduced below). Specifically, for the Incremental Pruning Ratio strategy, both logarithmic and linear approaches are employed. Each of these strategies offers distinct benefits: the linear approach achieves better Zero-shot Avg results but incurs a slight loss in PPL. Nevertheless, overall, whether the strategy is linear or logarithmic, the incremental scheme significantly outperforms uniform pruning or decrementing strategies.
>
> |                        | Model Size | PPL    | Zero-shot Avg. |
> | ---------------------- | ---------- | ------ | -------------- |
> | log increase (SlimGPT) | 3.40b      | 38.83  | 52.23          |
> | linear increase        | 3.34b      | 46.57  | 53.45          |
> | uniform                | 3.50b      | 123.05 | 44.34          |
> | log decrease           | 3.40b      | 380.69 | 36.73          |
> | linear decrease        | 3.34b      | 932.64 | 35.62          |
>
> Please note that our experiments are conducted under **low-resource conditions**. After extensive fine-tuning on large-scale data, the differences in performance resulting from various pruning ratio strategies will diminish further (Sheared-LLama[1], even with uniform pruning, mitigated performance impacts through subsequent training on a large data scale). However, under resource-limited conditions, selecting an appropriate pruning ratio strategy can reduce the reliance on subsequent training, thus minimizing performance loss. This is particularly crucial for LLMs, as a complete training cycle demands substantial resources and time.
>
> [1]. Sheared LLaMA: Accelerating Language Model Pre-training via Structured Pruning.
>
> > W2: Compare the throughput (eg., tokens per sec.) of pruned models generated by SlimGPT and the competing baselines.
>
> Thank you for your valuable suggestion. About the additional experiments on inference speed, we provide the experimental results and analysis in "global response" at the top. Please refer to that reponse for detailed information.
>
> > Q: Under the same total sparsity value, does different increasing schedules affect the model inference speedup?
>
> In general, under the same number of parameters, deeper models (with more layers) tend to have slower inference speeds. On the other hand, models with the same number of layers but different widths primarily experience variations in instantaneous computational load, which impacts speed differently depending on the performance and optimization of the GPU used.
>
> Since our current method typically does not affect the number of layers, as long as there isn't an extreme variation in width distribution, inference speed should not be significantly impacted. This is supported by the experimental results from the previous question.

---

> ### Author Response · Authors · 2024-08-13
>
> Dear Reviewer 8hG2:
>
> We sincerely appreciate your valuable and insightful comments. With less than 24 hours remaining in the discussion period, we anticipate further opinion from you.
>
> We would like to further discuss the LLM inference speed. Since most operators are computed in parallel, the acceleration from pruning does not arise from a reduction in computational load but rather from a decrease in parameter access time, which is a significant bottleneck in large model inference. Therefore, the inference time is not linearly related to the parameter number. In our experiments, we find that pruning 50% of the parameters result in an inference speed that is 63% of the original.
>
> Best Regards,
> Authors of submission 8375

---

### Official Review · Reviewer_QvFZ · 2024-07-09

**Soundness:** 2
**Presentation:** 3
**Contribution:** 2
**Rating:** 4
**Confidence:** 4

**Summary:**

This paper presents SlimGPT, a method for structured pruning of LLMs to balance performance with efficiency. The method is based on the OBS framework and introduces Batched Greedy Pruning to enhance pruning accuracy and speed. The authors also propose the Incremental Pruning Ratio strategy to mitigate performance loss due to error accumulation. Experimental results on LLaMA and other models demonstrate that SlimGPT achieves state-of-the-art results with significant improvements in performance retention compared to existing methods.

**Strengths:**

1. The paper introduces a novel method for structured pruning based on the OBS framework to accelerate large language models.
2. The method is validated through comprehensive experiments on various models, showing improved performance and efficiency.

**Weaknesses:**

1. Since Table 9 illustrates the significant impact of the calibration dataset on the pruning performance, I doubt whether the selection of calibration samples is more important than the design of the pruning criterion. The experimental results compared in the paper are with different sampling strategies of the calibration dataset, so it is hard to evaluate the superiority of the proposed method.
2. The design of the layer-wise sparsity is empirical with no theoretical analysis. Since the pruning within a layer is a greedy pipeline, it is unclear why the layer-wise sparsity design is not in a greedy pipeline.
3. The inference speed should be provided for comparisons.

**Questions:**

1. In Figure 1., from my view, it seems that the output elements of the third head are all the smallest, so they are all reordered to the first head for pruning. My question is what if not all elements of a head are in the same magnitude order? In this case, how to conduct batched greedy pruning?
2. Can you provide some insights about the reason why the pruning results with 2048 samples and 2048 sequence length start to decrease in zero-shot average metric?

**Limitations:**

Not applicable.

---

> ### Author Rebuttal · Authors · 2024-08-06
>
> We truly appreciate the reviewer for the constructive comments.
>
> > W1: The impact of the calibration dataset on the pruning performance.
>
> Thank you for the valuable comment. Table 9 displays the pruning results of SlimGPT **without fine-tuning** (for convenience, Table 9 is reproduced below). When the calibration data is switched from C4 to Alpaca/GPT4-Alpaca, the PPL results indeed degrade (48.26/47.06 vs. 42.06). However, the Zero-shot Avg scores improve significantly (54.44/54.66 vs. 52.70), surpassing the **fine-tuned results** of all baselines.
>
> | Dataset      | PPL↓  | Zero-shot Avg.↑ |
> | ------------ | ----- | --------------- |
> | C4 (SlimGPT) | 42.06 | 52.70           |
> | Alpaca       | 48.26 | 54.44           |
> | GPT4-Alpaca  | 47.06 | 54.66           |
>
>
> This observation highlights a trade-off between PPL and Zero-shot Avg. Due to SlimGPT's inherent parameter compensation mechanism, it is sensitive to the quality of the input data, so there is a distinct difference in impact between pre-trained data and instruction-following data. Additionally, our experiments show that random open-source pre-trained data can achieve SOTA results. Therefore, we believe that using higher-quality data tailored to specific domains provides SlimGPT with greater potential for improvement compared to other methods.
>
> > W2: The design of the layer-wise sparsity is empirical.
>
> The core concept of SlimGPT is derived from the OBS framework, which addresses global model pruning by breaking it down into layer-wise subproblems. Each layer is optimized **sequentially** from shallow to deep. Previous studies, such as OBC[1] and GPTQ[2], have demonstrated that this method yields excellent results across various domains. And our primary objective is to apply this framework to the structured pruning of LLMs.
>
> We would like to discuss the viability of the layer-wise greedy pipeline. Due to the unidirectional influence between layers, the current layer is impacted solely by the preceding layer and remains unaffected by subsequent layers. If the pruning process is not executed sequentially, the local optimality at each step would be compromised. We believe this would complicate the task and significantly increase the computational time required.
>
> [1]. Optimal Brain Compression: A Framework for Accurate Post-Training Quantization and Pruning.
> [2]. GPTQ: Accurate Post-training Quantization of Generative Pretrained Transformers.
>
> > W3: The inference speed should be provided.
>
> Thank you for your valuable suggestion. About the additional experiments on inference speed, we provide the experimental results and analysis in "global response" at the top. Please refer to that reponse for detailed information.
>
> > Q1: Further explanation on Figure 1.
>
> As a structured pruning scheme, SlimGPT prunes attention blocks with the smallest pruning granularity at the head level. Consequently, heads are treated as indivisible units. We evaluate each head by summing the errors of all columns within it. For those interested in the specifics, our detailed algorithmic process is outlined in Algorithm 1.
>
> > Q2: Can you provide some insights about the reason why the pruning results with 2048 samples and 2048 sequence length start to decrease in zero-shot average metric?
>
> Thank you for the comment. The Zero-shot Avg is essentially a mean value, which can be easily influenced by sub-tasks with significant fluctuations. To facilitate analysis, we provide detailed results for the commonsense reasoning tasks below.
>
> | Experiment | Sample Size | Sequence Length | BoolQ | PIQA  | HellasW | WinoG | ARC-e | ARC-c | OBQA  | Avg.  |
> | ---------- | ----------- | --------------- | ----- | ----- | ------- | ----- | ----- | ----- | ----- | ----- |
> | (a)        | 512         | 64              | 63.12 | 68.82 | 51.34   | 57.62 | 46.17 | 29.35 | 35.00 | 50.20 |
> |            | 512         | 1024            | 67.22 | 71.06 | 55.23   | 59.27 | 53.45 | 31.14 | 36.60 | 53.42 |
> |            | 512         | 2048            | 66.73 | 71.60 | 55.35   | 57.30 | 53.11 | 32.42 | 35.60 | 53.16 |
> | (b)        | 128         | 512             | 65.66 | 69.37 | 54.07   | 58.33 | 50.04 | 30.55 | 35.2  | 51.89 |
> |  | 256        | 512         | 63.76           | 70.29 | 54.47 | 58.88   | 53.07 | 30.89 | 34.6  | 52.28 |
> |            | 1024        | 512             | 65.60 | 71.65 | 54.35   | 57.46 | 53.07 | 30.63 | 35.40 | 52.59 |
> |            | 2048        | 512             | 63.30 | 71.71 | 54.79   | 57.38 | 53.2  | 31.66 | 34.60 | 52.38 |
>
> As the sample length increases from 64 to 1024, there is a noticeable improvement in the metrics across various tasks. However, when the length is further increased to 2048, the rate of improvement slows down, and some metrics even decline, particularly for WinoGrande (59.27 vs. 57.3). We believe that since most commonsense reasoning tasks involve short text data, the impact of SlimGPT on these tasks may diminish when the sequence length of the calibration set exceeds a certain threshold. In fact, it could even weaken the model's understanding of short texts.
>
> When the sample size increases, the changes across the various subtasks are less consistent and exhibit smaller magnitudes. As shown in Figure 3 of the paper, the fluctuations in Zero-shot Avg in subplot (a) are significantly smaller than those in subplot (b), and there are two instances of decline. Both of these declines result from fluctuations in the BoolQ dataset (refer to the table above). Thus, we hypothesize that the BoolQ dataset is particularly sensitive to random sampling in the calibration set, which may result in the drop in Zero-shot Avg.

---

> ### Author Response · Authors · 2024-08-13
>
> Dear Reviewer QvFZ:
>
> As the discussion period comes to a close, we sincerely look forward to your feedback on our rebuttal. Your further opinions would be essential for us to improve our work.
>
> Regarding your concern that the calibration set is more important than the pruning method, we include additional experiments with LLM-Pruner (our baseline). We perform pruning without fine-tuning at the same pruning ratio, yielding an evaluation result of `PPL=136.19`. In contrast, the worst result for SlimGPT is `PPL=48.26`. Therefore, we believe that while the calibration set may have a slight influence on the bias of model pruning, it does not fundamentally affect the pruning results.
>
> Best Regards,
> Authors of submission 8375

---

### Official Review · Reviewer_jqTK · 2024-07-13

**Soundness:** 2
**Presentation:** 3
**Contribution:** 3
**Rating:** 6
**Confidence:** 5

**Summary:**

The authors proposed a layer-wise pruning approach called SlimGPT that follows the Optimal Brain Surgeon framework but with a batched pruning procedure utilized to make it feasible on large models while remaining structured. The authors claim near-optimal pruning performance on commonsense reasoning datasets and wikitext ppl.

**Strengths:**

Models pruned via structured pruning approaches can naturally gain efficiency benefits, and the proposed method supposedly inherits this important property (though missing some efficiency reports). The task- and relatively architecture-agnostic nature of SlimGPT also makes it score well in terms of adaptability. The experiment reports indicate SlimGPT beats three other SOTA methods by a healthy margin (though its alignment requires some additional polish).

**Weaknesses:**

The main weakness of the paper is its eval, both in terms of alignment and coverage.

1. Unaligned evaluation: Many of the compared baselines utilized a different calibration dataset and procedure, but it looks like only the LLM-Pruner is replicated in an aligned setup, while all results for other methods are copied from their original literature. This needs to be controlled, especially with only three methods to compare.

2. Overly emphasis on the Llama family: The presented evaluation is solely conducted on various Llama 1/2 and Vicuna models, which are all Llama family-based. More coverage on other popular LLMs should be included. I'd recommend a healthy selection from Mistral, Phi, Gemma, Qiwen, and something along the line.

3. Only on zero-shot commonsense reasoning tasks: As mentioned around line 243, the real task evaluation of this paper is conducted *"under a zero-shot setting on the Commonsense Reasoning datasets, which encompass seven diverse subtasks..."* This is not comprehensive enough. Common intelligence datasets like MMLU and GSM8k in typical few-shot setups should also be reported. Additionally, given the weak generation/long context performance in some recent layer pruning (but not layer-wise structured pruning in finer granularity) work like ShortGPT, I'd like to see SlimGPT evaluated on tasks like LongBench, InfinityBench, Needle-In-A-Haystack retrieval, and HumanEval on models with longer context window (e.g., mistral 32k).

4. Incomplete efficiency report: There are no throughput or latency results, which are key efficiency metrics for conducting structured pruning in the first place and must be reported in efficiency literature, especially because different structured pruning methods can reflect different inference efficiencies, leading to work like ShearedLlaMA proposing targeted structured pruning. Also, the authors claim that *"low-cost, low-resource, and time-efficient compression scheme"* as their contribution in line 65, but there is no runtime or memory report on its pruning procedure.

5. Not really a weakness, but the authors should consider giving a more detailed introduction of the compared baselines either in related work or around line 246.

Despite my score of 4, I believe many of the raised concerns are addressable as there are mostly just more evals, and I am open to improving my rating upon proper rebuttal.

**Questions:**

1. Where is the plotting for the unpruned model in Figure 2? What pruning method is applied here? How much performance can fine-tuning recover in this setup?

2. Is a pruned model superior to a smaller pretrained model? Like can a SlimGPT-pruned 13B model with a pruning ratio set around 45% be more performant than a 7/8B model? From the look of Tables 1 & 2, it seems a 50% pruned 13B model is significantly inferior to a 7B, and a 50% pruned 30B model is required to provide similar zero-shot task performance to an unpruned 7B, so this is unlikely. If confirmed, I am afraid this massively discounts the contribution of this work, as one can often just adopt a smaller pretrained model with no pruning or calibration necessary. Though I understand different application scenarios may call for models with different sizes, where the pretrained models can't cover them all.

**Limitations:**

The authors have provided a limitation and checklist section.

---

> ### Author Rebuttal · Authors · 2024-08-06
>
> We truly appreciate the reviewer for the constructive comments. Due to text limitations, we try to answer your questions concisely and convincingly.
>
> > W1: Unaligned evaluation.
>
> We acknowledge your concerns. Achieving a fully aligned experimental setup is challenging since pruning tasks differ from tasks like model optimization, and various pruning methods have unique principles and data requirements. For instance, `LLM-Pruner` has two stages: pruning and fine-tuning, while `Compresso` and `LoRAPrune` involve only one stage of sparse training. Our method, `SlimGPT`, also follows a two-stage process, aligning it with `LLM-Pruner`. Given the differing principles of `Compresso` and `LoRAPrune`, we make a compromise in terms of aligning the experimental environment. However, we believe that this does not affect the conclusions of our paper. In fact, we achieve better pruning results using fewer data resources or fewer iterations.
>
> > W2: Overly emphasis on the Llama family.
>
> Thank you for your valuable comments. In our model selection, we reference previous works and the influence of LLama and Vicuna models but overlooked Vicuna's derivation from LLama. Due to time constraints, we choose Baichuan-7b, a prominent model in the Chinese community, for our experiments. This model includes a comprehensive MMLU and C-Eval evaluation script and is supported by LLM-Pruner, aiding our quick verification process.
>
> Using the same setup as the paper, we prune Baichuan-7b by 20% with LLM-Pruner and SlimGPT. SlimGPT achieves slightly better PPL results (19.73 vs. 19.85) and significantly outperforms LLM-Pruner in all commonsense reasoning tasks (58.8 vs. 56.38).
>
> Prune%|Method|#Params|PPL↓|BoolQ|PIQA|HellasW|WinoG|ARC-e|ARC-c|OBQA|Avg
> -|-|-|-|-|-|-|-|-|-|-|-
> -|-|7B|13.25|70.09|76.93|70.06|64.09|67.05|40.53|38.20|60.99
> 20%|LLM-Pruner|5.7B|19.85|62.87|74.48|63.03|60.93|60.31|36.86|36.20|56.38
> 20%|SlimGPT w/o tune|5.7B|20.01|69.17|75.03|65.25|61.25|64.94|35.15|36.60|58.20
> 20%|SlimGPT|5.7B|19.73|66.21|75.03|66.74|63.85|62.84|38.91|38.00|58.80
>
> > W3: Only on zero-shot commonsense reasoning tasks.
>
> Thank you for your suggestions. We evaluated our model based on the official LLama report and previous research, focusing on language modeling performance (PPL) and zero-shot commonsense reasoning. While we believe these tasks are convincing, we recognize they may be insufficient for a complete assessment. To improve our evaluation, we perform 5-shot tests on Baichuan-7b using MMLU and cross-lingual task C-Eval, as seen in the table below.
>
> The 5-shot evaluation results for MMLU show SlimGPT outperforming the baseline (35.4 vs. 24.3), and it also excels in the cross-lingual C-Eval dataset (28.7 vs. 22.4), despite reduced performance after finetuning on Alpaca.
>
> Dataset|Prune%|Method|#Params|Humanities|Social Sciences|STEM|Other|Avg
> -|-|-|-|-|-|-|-|-
> MMLU|-|-|7B|38.1|49.2|35.2|47.7|42.1
> ||20%|LLM-Pruner|5.7B|24.8|23.4|21.9|26.8|24.3
> ||20%|SlimGPT w/o tune|5.7B|29.7|38.0|31.3|38.9|34.0
> ||20%|SlimGPT|5.7B|32.2|39.2|31.1|40.2|35.4
> C-Eval|-|-|7B|47.2|50.5|36.4|45.5|43.3
> ||20%|LLM-Pruner|5.7B|23.5|24.9|21.1|21.2|22.4
> ||20%|SlimGPT w/o tune|5.7B|32.9|34.3|28.3|30.9|31.0
> ||20%|SlimGPT|5.7B|31.8|33.1|24.2|29.9|28.7
>
> Regarding your suggestion for experiments on long-context performance, we recognize their importance but, due to time constraints, we won’t be able to conduct them at this moment. We plan to include them in future work for a more comprehensive study.
>
> > W4: Incomplete efficiency report.
>
> Thank you for your suggestion. We include the experimental results and analysis on inference speed and pruning efficiency in the "global response" section, as other reviewers have similar questions. Please refer to that for detailed information.
>
> > W5: Detailed introduction of the compared baselines.
>
> Due to space constraints, we have omitted the baseline introduction in the final paper version. We apologize for any inconvenience and will include it in the updated version.
>
> > Q1: Further explanation on Figure 2.
>
> Figure 2 shows the output errors of various pruned models relative to the unpruned model, which has an error of zero and is not displayed (its PPL is 12.63). In our pruning approach, we utilize SlimGPT with the Alpaca dataset, pruning only the first layer to minimize output errors.
>
> To assess performance recovery after finetuning, we present the PPL results below, where all models are finetuned with LoRA for one epoch using Alpaca. We observe that the PPL does not improve after finetuning, even for the unpruned model. We believe this may be due to the distinct distribution bwtween the instruction-following dataset Alpaca and test dataset Wikitext2, potentially causing overfitting of the unpruned weights and poorer performance on the test data, which requires further validation.
>
> Model|PPL (w/o tune)|PPL (w/ tune)
> -|-|-
> LLama-7B|12.63|15.63
> Layer0-prune-25%|12.86|16.86
> Layer0-prune-50%|13.98|17.31
> Layer0-prune-75%|21.49|31.27
>
> >  Q2: Is a pruned model superior to a smaller pretrained model?
>
> This topic deserves discussion. Under **low-resource conditions**,  pruned models after finetuning generally perform worse than smaller pre-trained ones, as shown in our works and previous research(LLM-Pruner). However, with **full training**, pruned models can outperform their smaller versions, like in Sheared-LLama[1]. Thus, pruning may serve as a high-benchmark initialization method to lessen the need for extensive training.
>
> Our aim is to focus specifically on the task of LLM pruning itself. Under constrained resource conditions, such as when a more compact version of the model (e.g., less than 1B parameters) is required for edge-side deployment, LLM pruning and compression provide a cost-effective solution.
>
> [1]. Sheared LLaMA: Accelerating Language Model Pre-training via Structured Pruning.

---

> ### Comment · Reviewer_jqTK · 2024-08-12
> **Bumping the score to 5, but would need more evals and comparsions to keep increasing my rating.**
>
> I thank the authors for the detailed rebuttal. The new results — especially on the efficiency front — look decent and thus warrant a slight bump to 5. However, to fully convince me, I'd like to see SlimGPT applied on truly challenging tasks like GSM8k, as well as long context tasks like LongBench and Needle-in-a-Haystack (on llama3 and mistral v0.2). I am particularly interested in the long context front due to the known drawbacks of ShortGPT.
>
> With the unaligned nature discussed in W1, it looks like the proposed method is mostly compared to LLMPruner. While I recognize that LLMPruner is an established method, I wonder how SlimGPT would perform against some of the more advanced methods, like APT [2].
>
> p.s. While I appreciate the addition of Baichuan, I would still like to see the MMLU report on a more mainstream model, like llama2-7b, for better cross-referencing needs. May the authors supply that?
>
> ---
>
> [2] APT: Adaptive Pruning and Tuning Pretrained Language Models for Efficient Training and Inference

---

> > ### Author Response · Authors · 2024-08-13
> >
> > Dear Reviewer jqTK:
> >
> > We truly appreciate the reviewer's recognition of our work. In response to your suggestions regarding extra experiments, we add the following evaluations:
> >
> > > Evaluation of LLama2-7b on MMLU and mathematical tasks GSM8k.
> >
> > | Prune% | Method           | #Params | Humanities | Social Sciences | STEM | Other | Avg  | GSM-8k-Acc  |
> > | ------ | ---------------- | ------- | ---------- | --------------- | ---- | ----- | ---- | ---- |
> > | -      | -                | 6.7B    | 43.3       | 51.6            | 36.3 | 52.1  | 45.6 | 13.8 |
> > | 20%    | LLM-Pruner       | 5.4B    | 25.7       | 23.6            | 24.2 | 26.8  | 25.2 | 2.3  |
> > | 20%    | SlimGPT w/o tune | 5.4B    | 36.0       | 45.2            | 33.5 | 44.1  | 39.4 | 4.2  |
> > | 20%    | SlimGPT          | 5.4B    | 35.3       | 42.2            | 31.5 | 43.0  | 37.8 | 6.0  |
> >
> > The first few columns display the results for MMLU, while the last column shows the evaluation results for GSM-8k. SlimGPT demonstrates significant improvements over the baseline in both tasks. Notably, for the challenging task of GSM-8k, LLM-Pruner retains 16.7% of the performance (2.3 vs. 13.8) after pruning, whereas SlimGPT retains 43% of the performance (6.0 vs. 13.8), achieving more than a twofold increase. Since we employ a very basic and low-cost fine-tuning method, we believe there is room for further performance enhancement.
> >
> > > Evaluation of Mistral-7B-Instruct-V2.0 on LongBench
> >
> > Due to the more representative 32k context of Mistral, we will present the evaluation results for Mistral-7B-Instruct-V2.0 on LongBench, hoping this will serve as a valuable reference.
> >
> > The model evaluation takes longer than we anticipate, so we primarily provide results for the single-document QA task, as shown in the table below. The performance of the pruned model varies on English tasks, with the capability retaining as much as 97% (30.54 vs. 31.47) in some instances. However, its performance on the cross-language task MultiFieldQA-zh is comparatively modest. We find that this can be largely attributed to the Alpaca fine-tuning, which results in a bias towards answering questions in English, consequently lowering the overall scores. For example, consider the following case.
> >
> > `{'pred': 'The appeals court determined that the defendant should pay a compensation amount of RMB 57,081.86.', 'answers': ['人民币57081.86元。'], 'all_classes': None, 'length': 2975}`
> >
> > | Prune% | Method  | #Params | NarrativeQA | Qasper | MultiFieldQA-en | MultiFieldQA-zh |
> > | ------ | ------- | ------- | ----------- | ------ | --------------- | --------------- |
> > | -      | -       | 6.7B    | 27.33       | 31.47  | 48.59           | 58.17           |
> > | 20%    | SlimGPT | 5.4B    | 18.89       | 30.54  | 40.47           | 24.75           |
> >
> >
> > > SlimGPT vs. APT.
> >
> > The APT paper actually describes a sparse training pruning method. This approach manually inserts masks before and after specific modules and measures the importance of channels/heads using outliers. Its advantage lies in an end-to-end design; however, its task-specific nature and strong coupling with LoRA complicate its application to other tasks.
> >
> > Based on the results provided in the paper, we conduct a rough comparative analysis. While the different pruning ratios make it difficult to directly judge the evaluation results, it is evident that SlimGPT, with 1 epoch naive LORA tuning, offers a more lightweight and straightforward setup without requiring alterations to the model structure.
> >
> > | Method  | Prune% | Tuning Data | Tuning Epoch | LORA rank | HellaSwag Eval | MMLU Eval |
> > | ------- | ------ | ----------- | ------------ | --------- | -------------- | --------- |
> > | APT     | _30%_    | Alpaca      | 15           | 8-256     | 71.1           | 36.9      |
> > | SlimGPT | _20%_    | Alpaca      | **1**            | **8**         | 74.9           | 37.8      |
> >
> >
> > Best Regards,
> > Authors of submission 8375

---

> ### Comment · Reviewer_jqTK · 2024-08-13
> **Why can't you prune SlimGPT 30% to be comparable with your APT report?**
>
> I'll take a closer look at the rest soon, but may the author please address the question as titled? Just want to post earlier so that you can have a chance to reply. I'd also like to see APT on GSM8k under a comparable setting.

---

> > ### Author Response · Authors · 2024-08-14
> >
> > Dear Reviewer jqTK:
> >
> > We would like to thank the reviewer for the comments.
> >
> > > Why can't you prune SlimGPT 30% to be comparable with your APT report?
> >
> > Our analysis is based on the results provided in the APT paper and the existing results from the SlimGPT paper. Unfortunately, we do not have sufficient time to conduct additional experiments to prune the model by 30% for comparison now (The experiment is ongoing). We hope you can understand this limitation. Furthermore, the APT paper do not provide evaluation results for GSM8K, so we are currently unable to present GSM8K comparison results.
> >
> > > More evaluation of long-context tasks on LongBench.
> >
> > We update our evaluation of long-context tasks on LongBench, which includes 4 task types, as shown in the table below. The current version of the model evaluation has fixed bugs related to input format compared to the previous version, leading to a slight change in performance.
> >
> > According to the table, we find that the performance of the pruned model varies on **English** tasks; in some cases, it even exceeds the results of the original model (2WikiMQA: 28.34 vs. 26.32). The majority of tasks maintain over 80% of the original performance.
> >
> > - Evaluations on Single-Doc QA tasks
> >
> > | Prune% | Method           | #Params | NarrativeQA | Qasper | MultiFieldQA-en | MultiFieldQA-zh |
> > | ------ | ---------------- | ------- | ----------- | ------ | --------------- | --------------- |
> > | -      | -                | 6.7B    | 27.32       | 31.47  | 48.57           | 49.06           |
> > | 20%    | SlimGPT          | 5.4B    | 19.33       | 30.37  | 42.64           | 26.65           |
> >
> > - Evaluations on Multi-Doc QA tasks
> >
> > | Prune% | Method           | #Params | HotpotQA | 2WikiMQA|Musique|DuReader (zh)|
> > | ------ | ---------------- | ------- | ----------- | ------ | --------------- | --------------- |
> > | -      | -                | 6.7B    | 43.11       | 26.32  | 18.81           | 30.57           |
> > | 20%    | SlimGPT          | 5.4B    | 38.13       | 28.34  | 15.16           | 13.98           |
> >
> > - Evaluations on Summarization tasks
> >
> > | Prune% | Method           | #Params |GovReport|QMSum|MultiNews|VCSUM (zh)|
> > | ------ | ---------------- | ------- | ----------- | ------ | --------------- | --------------- |
> > | -      | -                | 6.7B    | -           | 22.92  | 25.46           | 14.91           |
> > | 20%    | SlimGPT          | 5.4B    | -           | 19.95  | 22.78           | 12.49           |
> >
> > - Evaluations on Few-shot Learning tasks
> >
> > | Prune% | Method           | #Params |TREC|TriviaQA|SAMSum|LSHT (zh)|
> > | ------ | ---------------- | ------- | ----------- | ------ | --------------- | --------------- |
> > | -      | -                | 6.7B    | 68.50       | 86.98  | 42.00           | 39.00           |
> > | 20%    | SlimGPT          | 5.4B    | 56.00       | 82.59  | 41.46           | 22.75           |
> >
> > Best Regards,
> > Authors of submission 8375

---

> > > ### Comment · Reviewer_jqTK · 2024-08-14
> > > **What's slowing things down from obtaining a 30% pruned SlimGPT?**
> > >
> > > Per your global rebuttal, pruning a Llama2-7B to 50% only takes a little over 1,000 seconds, which is almost a trivial cost, and a few epochs of Alpaca are often pretty fast too. So what is actually slowing things down to the point that you *"do not have sufficient time to conduct additional experiments to prune the model by 30% for comparison now (The experiment is ongoing)"*?
> > >
> > > Just trying to make sure I am not missing any major things.

---

> > > > ### Author Response · Authors · 2024-08-14
> > > >
> > > > Dear Reviewer jqTK:
> > > >
> > > > In fact, all of our hardware resources were dedicated to running the LongBench experiments, which exceeded our expectations in terms of resource demands. Just before our previous response, we completed the LongBench experiments. Now we finish the pruning experiment for LLaMA2-7B at 30% and the subsequent evaluation. Below, we present our experimental results (with APT data referenced from the paper). We primarily focus on comparing the HellaSwag and MMLU tasks, as these two datasets are used for evaluation in both papers. We train for 3 epochs during fine-tuning. SlimGPT performs slightly lower than APT on MMLU but demonstrates better performance on HellaSwag. And it is important to note that SlimGPT requires fewer iterations and less trainable parameters for LoRA during the fine-tuning phase.
> > > >
> > > > | Method  | Prune% | Tuning Data | Tuning Epoch | LORA rank | HellaSwag Eval | MMLU Eval |
> > > > | ------- | ------ | ----------- | ------------ | --------- | -------------- | --------- |
> > > > | APT     | **30%**    | Alpaca      | 15           | 8-256     | 71.10           | 36.9      |
> > > > | SlimGPT | **30%**    | Alpaca      | 3            | 8         | 72.42           | 35.2      |
> > > >
> > > > Best Regards,
> > > > Authors of submission 8375

---

> ### Comment · Reviewer_jqTK · 2024-08-14
> **Thanks. Bumping to 6 but please include the additional results, as well as tone down your claim a little.**
>
> Thank you for being resourceful and adding many requested experiments during the rebuttal time. **The added result confirms my intuition: that cheap, non-ShreadLlama-like LLM pruning techniques do not perform well under rigorous evaluation.** Your added results on GSM8k and LongBench confirm that, as there are visible drops with just 20% pruned. Note that we usually don't observe such a performance drop with techniques like weight-only quantization at a much more aggressive rate; even with vanilla group-wise quantization with no finetuning.
>
> That being said, I recognize that pruning LLMs is much harder than quantifying LLMs. There surely are some benefits unique to the pruning way, and overall, pruning is without a doubt a school of efficiency worth developing; especially knowing its gap with quantization. **The proposed method is better or at least on par with the established/recent baselines, so I recommend an acceptance with score 6 & confidence 5.** But I urge the authors to:
>
> * Tone down the claim a bit, e.g., the "98% performance" claim in your abstract is slightly misleading. Most common-sense reasoning tasks are easy and do not represent LLM's true capability, so it is almost an overstatement based on cherry-picked results.
> * Highlight the results that are not perfect (MMLU, GSM8k, LongBench, etc.) so that future works will have a clear direction for improvement, instead of always muddling those easy tasks.
> * Add a proper section to discuss the pros/cons of pruning compared to other efficiency techniques (e.g., quantization) and their unique challenges.

---

> > ### Author Response · Authors · 2024-08-14
> >
> > Dear Reviewer jqTK:
> >
> > Thanks for your valuable feedback. We sincerely appreciate your time to review our submission and response. We will revise the paper accordingly and incorporate the above results into the updated version.
> >
> > Best Regards,
> > Authors of submission 8375

---

### Author Rebuttal · Authors · 2024-08-06

Dear Reviewers,

We sincerely appreciate your valuable and insightful comments. Here I would like to address the concerns regarding inference speed or  pruning efficiency raised by all reviewers.

> Inference speed and memory usage report.

As the inference speed is primarily influenced by the final model structure and is not specifically tied to the pruning algorithm used (typically, the number of layers does not decrease), we initially omitted the inference runtime report. To demonstrate that SlimGPT actually helps to deploy LLMs, we provide the inference speeds for LLama-7b with 20% and 50% pruning, as shown in the table below. The batch size is set to 1, the maximum output limit is 512, and the average value is taken from 50 inference results. Additionally, we examine the impact of two different pruning ratio strategies on inference speed: uniform pruning and the Incremental Pruning Ratio strategy. All supplementary experiments were conducted in an environment with NVIDIA H20.

In the case of pruning 50% of the parameters using the log increase strategy, the model's memory usage during inference is reduced to 51% (14297MB vs. 27737MB), and the inference latency decreases to 63% (9.21ms vs. 13.51ms). When employing uniform pruning, both memory usage and latency experience slight reductions, although it is important to note that the parameter counts are not entirely equivalent between the two methods.

| Prune% | Strategy     | #Params | Memory  | Avg Latency (per token) |
| ------ | ------------ | ------- | ------- | ----------------------- |
| -      | -            | 6.7B    | 27737MB | 13.51ms                 |
| 20%    | log-increase | 5.4B    | 22497MB | 11.89ms                 |
| 50%    | log-increase | 3.4B    | 14297MB | 9.21ms                  |
| 50%    | uniform      | 3.4B    | 13793MB | 9.05ms                  |

> Pruning runtime and memory usage report.

Regarding the runtime and memory usage during the pruning procedure, we have mentioned briefly that all pruning processes can be completed within 1 GPU hour (using A100 hardware). Specifically, the memory usage varies depending on the model size and the calibration size, and the pruning speed is additionally influenced by the pruning ratio. We present the pruning efficiency results from our paper experimental setup.

Since SlimGPT operates in a layer-wise manner, we don't need to load the entire model but only load the parameters of the current layer and the corresponding input features at one time, which significantly reduces memory usage. For the task of pruning the 13B model by 50%, we only require 12 GB of GPU memory and 41 minutes to complete the process.

| Model     | memory | prune-20%-runtime | prune-50%-runtime |
| --------- | ------ | ----------------- | ----------------- |
| LLama-7b  | 7375M  | 678.4s            | 1073.9s           |
| LLama-13b | 11601M | 1417.1s           | 2475.3s           |

---

### Decision · Program_Chairs · 2024-09-25

**Decision:**

Accept (poster)

**Comment:**

The authors introduced a low-cost and fast structured pruning method named SlimGPT. Experiments results show that SlimGPT outperforms other methods and achieves state-of-the-art performances.

We appreciate the responses from authors and the discussions with reviewers. Please add all the mentioned experiments and analysis into the final version.